# Ecological Sustainability and Households’ Wellbeing: Linking Households’ Non-Traditional Fuel Choices with Reduced Depression in Rural China

**DOI:** 10.3390/ijerph192315639

**Published:** 2022-11-24

**Authors:** Xiaoheng Zhang, Guiquan Yan, Qipei Feng, Amar Razzaq, Azhar Abbas

**Affiliations:** 1College of Economics and Management, Nanjing University of Aeronautics and Astronautics, Nanjing 211106, China; 2College of Economics and Management, Huazhong Agricultural University, Wuhan 430070, China; 3Business School, Huanggang Normal University, No. 146 Xingang 2nd Road, City Development Zone, Huanggang 438000, China; 4Institute of Agricultural and Resource Economics, University of Agriculture, Faisalabad 38040, Pakistan

**Keywords:** ecological support, human health, ESR model, green fuels, energy transition, environmental management

## Abstract

A sustainable and pleasant environment is deemed to offer various positive externalities such as scenic, visual and behavioral archetypes and patterns exhibiting in various forms. Such a scenario can significantly relieve households from many psychological and personal complications such as depression. Depression has aroused great concerns in recent years due to its personal and social burdens and unforeseeable damage. Many studies have explored the effects of air pollution caused by traditional fuel consumption on depression. However, limited evidence is available on how household non-traditional fuel choices affect depression. Based on a nationally representative dataset collected from China Family Panel Studies (CFPS) in 2012, this paper employs an endogenous switching regression (ESR) model and an endogenous switching probit (ESP) model to address the endogenous issue and to estimate the treatment effects of non-traditional fuel choices on depression in rural China. The empirical results show that non-traditional fuel users have significantly lower Epidemiologic Studies Depression Scale (CES-D) scores, indicating non-traditional fuel users face a lower risk of depression. Compared to solid fuels, employing non-traditional fuels will lead to a 3.659 reduction in depression score or decrease the probability of depression by 8.2%. In addition, the results of the mechanism analysis show that household non-traditional fuel choices affect depression by reducing the probability of physical discomfort and chronic disease. This study provides new insight into understanding the impact of air pollution in the house on depression and how to avoid the risk of depression in rural China effectively.

## 1. Introduction

Depression has become a common mental health problem in recent years, resulting in adverse effects such as a high risk of disability and even mortality [1]. At a global level, over 4.4% of the world’s population is estimated to suffer from depression, and the number is still growing [2]. Depressive illness is the second leading cause of physical and mental disability and takes the largest share of the world’s total costs to address disease [3,4,5]. In China, the prevalence of depression is 3.02%, which is lower than the world’s average, and depression has risen from the 15th leading cause of all-cause disability-adjusted life years in 1990 to the 10th in 2017 in China [6]. However, we may underestimate the actual rate of depression because of stigma and the lack of standard diagnostic criteria in China [4]. Evidence confirms the underestimation and shows that the prevalence of depression in the rural Chinese population reaches 5.9%, higher than the world’s average [7]. Yet we lack effective therapies and mental-health resources to treat depression. Thus, the substantial personal and social burdens and unforeseeable damage from depression have caused widespread concerns. It is urgent to identify the determinants of depression and to avoid the risk effectively.

Depression is associated with multiple risk factors, including genetic profiles, socioeconomic status, lifestyle, and air pollution [1,8]. Many studies have linked depression with low socioeconomic status and support, poor health status, and female gender [9,10,11]. Some scholars analyzed the data of about 1600 peasants in rural China and concluded that the risk of depression arose from low family income and other indicators of social position. They also found that high levels of social support in the community might contribute to a lower risk of depression [12]. The prevalence of depression is also significantly associated with rural-urban inequalities in educational level and income [13]. Moreover, studies have also revealed that environmental pollution can result in depression symptoms, and increased access to greenness contributes to a lower risk of depression and anxiety. Therefore, environmental pollution can act as valuable evidence for the growing prevalence of depression worldwide [1,14,15].

Previous research showed that the increased ambient air pollutants would significantly negatively influence mental health. With the increased consumption of traditional fuels driven by rapid industrialization and urbanization over the past decades, air pollution, especially airborne particulate matter (PM) pollution, has become a severe environmental problem in China, especially in urban areas [16]. Exposure to PM 2.5, or a higher concentration of PM10, nitrogen dioxide (NO2) and ozone may increase the risk of depression [17,18]. A biological explanation for the impacts of air pollutants on depression is that it results in increased oxidative stress, cerebrovascular damage, neurodegeneration or neuroinflammation [1,19]. Some scholars also argued that ambient air pollutants prevent people with chronic disease from outdoor physical activities, and hereafter the aggravation of their disease condition leads to or worsens depression [20,21]. In addition, bad air quality may lead to depression by directly affecting their physical and social activities, thus lowering hedonic happiness [22,23].

Limited evidence is available on how household non-traditional fuel choices affect depression, while a large number of studies have explored the effects of traditional fuel consumption on depression through outdoor air pollution. The burning of conventional fuels, such as coal and oil, is the major contributor to the increased particular matter in the outdoor air. Solid fuels such as charcoal, wood, and crop residues are the major household energy source for more than 700 million people in rural China [1]. While most urban areas have achieved full coverage of natural gas, traditional solid fuels are still widely used for cooking and heating in rural areas because of poor economic conditions and conventional lifestyles. The burning of solid fuels produces numerous toxic air pollutants, such as particles, nitrogen oxide, carbon monoxide, and organic air pollutants, and leads to higher levels of household air pollution due to incomplete combustion and lack of ventilation facilities [24]. Household air pollution in the heating season is significantly higher than that in the non-heating seasons for indoor microenvironment categories in rural China [25]. Thus, solid fuels are an important source of household air pollution [26]. Similarly to outdoor air pollution, household air pollution would also result in health problems, such as hypertension, respiratory sleep problems, diabetic states, cardiovascular and respiratory diseases [16,27,28,29,30,31,32,33], and cause substantial disease burden [34].

Few studies have investigated the impact of household fuel use on depression. An early study conducted in rural India adopted randomized control trials and revealed that household air pollution from cooking using biomass would result in a higher prevalence of depression and deplete platelet serotonin for pre-menopausal women [35]. Two other studies conducted in China showed that long-term household air pollution exposure to solid fuels was associated with higher depression risk among middle-aged and older people in China [1,36]. Both of them used the data from the China Health and Retirement Longitudinal Study (CHARLS). The earlier one adopted the propensity score matching (PSM) method, considering the endogeneity issue, while the later one applied the Pearson test and survival analysis to evaluate the impacts. Given the lack of effective therapies to treat depression and the increasing concerns about depression, the main objective of this study is to examine to what extent household non-traditional fuel choices affect depression in rural China.

This paper contributes to the literature in three important ways. First, depression is more prevalent in rural than urban areas [37,38] and is often underestimated and misdiagnosed due to the lack of standard diagnostic criteria used to identify depression. Thus, we investigate the effect of household non-traditional fuel choices on depression in rural China. We believe this study will shed some light on the impact of non-traditional fuel uses on depression in the house and on how to effectively avoid the risk in rural China. Second, we use nationally representative individual-level data from China Family Panel Studies (CFPS) in 2012. It uses the Center for Epidemiologic Studies Depression Scale (CES-D) score developed by Radloff in 1977 to assess the level of depression that has been widely used in investigating depressive symptoms [39]. In 2012, the survey used a complete scale of 20 items, including somatic symptoms, depressed affect, positive affect and interpersonal problems, to measure the individual-level depression condition. Third, although using solid fuels may result in household air pollution, a large number of households in rural China have replaced solid fuels with non-traditional fuels, such as liquefied gas, natural gas, and electricity. The choice of solid or non-traditional fuels may not be randomly assigned and may be influenced by some unobservable factors (e.g., non-traditional fuel users have an innate preference for clean and neat living conditions). In addition, depression may also influence the choice of fuel use because people suffering from depression are more likely to choose non-traditional fuels which are convenient for collection and use. Thus, we adopt the widely used endogenous switching regression (ESR) model and endogenous switching probit (ESP) model to address the potential endogenous problem.

The remainder of this paper is organized as follows. Section 2 introduces methods and materials. Section 3 presents results and discussion.

## 2. Materials and Methods

### 2.1. Data and Variable Description

We collected the individual and household level data from the CFPS for the second wave in 2012 (website: https://opendata.pku.edu.cn/dataverse/CFPS (accessed on 20 August 2019)). It is a nationally representative and annual longitudinal survey launched in 2010 by the Institute of Social Science Survey (ISSS) of Peking University in China. The survey focuses on the economics, non-economic matters and well-being of the Chinese population. The CFPS is funded by the Chinese government. A multi-stage probability and implicit stratification procedure are used to determine the sample from the county, village and household stages for CFPS in 2012. The survey sample was drawn from 25 provinces covering 16,000 households, followed every two years. More detailed information can be found on the official website of CFPS.

There were 35,720 adults above the age of 16 interviewed in 2012. We dropped 9671 individuals who lived in urban areas, and 26,049 individuals were left. Then, we dropped 4725 individuals who were not living at home for the next three months. Finally, we obtained a sample of 21,324 individuals.

Table 1 shows the definitions and descriptive statistics of variables used in this paper. As shown in this table, the average depression score is 13.64, with higher scores implying greater depression severity. Depression status shows that 35% of residents in rural China suffer from depression. This value is significantly higher than the results reported in the literature [6,7]. The reported data for the prevalence of depression in the literature is the ratio diagnosed by doctors. If we replace the cutoff point of 16 with 28, which means strictly depression symptoms to define depression, the prevalence of depression is 6.43%. The share of non-traditional fuel adoption accounts for 51.5% of our sample. Table 1 also reports that the mean age of an individual is 45.85, with more than four persons in their family on average.

Table 2 presents the mean differences in characteristics between solid and non-traditional fuel users. The mean values of the variables have a statistical difference between them. We find significantly lower CES-D scores for non-traditional fuel users, indicating they face a lower risk of depression (Table A1 and Table A2 in Appendix A also report the mean differences by adjusting the *p*-value according to the sample size and the frequencies of the binary variables, respectively). Compared with solid fuel users, non-traditional fuel users tend to be younger, have better health status, smoke and exercise less frequently. While they have fewer household members, they have more assets per capita. However, simple descriptive statistics have no causal interpretation because they do not take confounding factors into consideration. Thus, we will employ the ESR model to estimate the treatment effects of the energy transition on depression in the next section.

### 2.2. Analytical Methods

In order to precisely investigate the impact of household non-traditional fuel choices on depression in rural China, we adopt the ESR model and the ESP model to address the potential endogenous problem. The ESR model is used to estimate the treatment effects when the outcome is a continuous variable (e.g., depression level in this paper) and the ESP model is employed as the outcome is a binary variable (e.g., depression status in this paper). As described previously, the empirical challenge for this paper is whether or not people choose to use non-traditional fuels is not randomly assigned. The fuel type choice is not only affected by observable factors (e.g., demographic and family characteristics) but also by unobservable factors such as the lack of access to non-traditional fuels or innate preference for clean living conditions supported by non-traditional fuel use. The unobservable factors will be omitted in the error term, leading to a correlation between it and the non-traditional fuel choice variable. In addition, depression and fuel choices have mutual causation because people suffering from depression may not have the ability to collect and use solid fuels. If an ordinary least squares (OLS) regression model is used to evaluate the effect, it will result in a biased estimation of energy transition effects on depression and misleading policy implications.

To the extent that non-traditional fuel choices are an endogenous variable, the study mentioned earlier has employed the PSM model to deal with the issue of endogeneity [1]. However, a well-known drawback of the PSM model is that it deals with the endogenous problem only caused by omitting observable factors rather than omitting unobservable factors [40]. We will use the ESR and ESP models, which are an instrumental variable-based approach that can address the endogenous problem by accounting for both observed and unobserved factors [41,42]. In addition, depression determinants for clean and solid fuel users are allowed to be different in the ESR and ESP models.

The ESR model was developed by Lee in 1982, and it is an upgrade form of Heckman’s selection approach [41]. The ESR model uses the full information maximum likelihood (FIML) estimation involving one selection and two outcome equations to evaluate the effects of solid fuel use. The ESR model (ESP model) proceeds in two stages, and we estimate a probit model in Equation (1) to identify the determinants of non-traditional fuel choices in the first stage.
(1)$$E_{i}^{*}=\gamma Z_{i}+\mu_{i},with\;\begin{array}{c} E_{i}=\left\{\begin{array}{c} 1,ifE_{i}^{*}>0\\ 0,otherwise \end{array}\right\} \end{array}$$
where $E_{i}$ is a binary indicator variable which takes the value of one if the individual’s household uses non-traditional fuels for cooking and heating, and zero otherwise. $Z_{i}$ is a vector of explanatory variables in this paper, including demographical characteristics (e.g., age, gender, religion, ethnicity, member of the Communist Party of China, health status, uncomfortable, chronic disease, hospital, smoking, medical insurance and exercise frequency) and household characteristics (e.g., asset per capita, family size); γ is a vector of parameters to be estimated; and $\mu_{i}$ is an error term which is assumed to be normally distributed with zero mean.

In the second stage, the determinants of depression level and depression status are regressed separately for solid and non-traditional fuel users, which are shown in Equations (2a) and (2b),

Regime 1 (solid fuel users)
(2a)$$D_{is}=\alpha_{is}X_{is}+\varepsilon_{is}\begin{array}{cc} if & E_{i}=1 \end{array}$$

Regime 2 (non-traditional fuel users)
(2b)$$D_{ic}=\alpha_{ic}X_{ic}+\varepsilon_{ic}\begin{array}{cc} if & E_{i}=0 \end{array}$$
where $D_{is}$ and $D_{ic}$ are depression scores (or depression status) measured by using CES-D for solid and non-traditional fuel users, respectively;$\;X_{is}$ and $X_{ic}$ refer to a vector of exogenous variables; $\alpha_{is}$ and $\alpha_{ic}$ are parameters to be estimated; $\varepsilon_{is}$ and $\varepsilon_{ic}$ are error terms.

The ESR model and ESP model calculates Inverse Mill Rations after estimating Equation (1) to deal with the endogenous issue resulting from unobservable factors. Thus, we add the Inverse Mill Rations to Equations (2a) and (2b), and rewrite them as follows:

Regime 1 (non-traditional fuel users)
(3a)$$D_{is}=\alpha_{is}X_{is}+\sigma_{\mu s}R_{is}+\theta_{is}\begin{array}{cc} if & E_{i}=1 \end{array}$$

Regime 2 (solid fuel users)
(3b)$$D_{ic}=\alpha_{ic}X_{ic}+\sigma_{\mu c}R_{ic}+\theta_{ic}\begin{array}{cc} if & E_{i}=0 \end{array}$$
where $\;D_{is}$,$\;D_{ic}$,$\;X_{is}$,$\;X_{ic}$,$\;\alpha_{is}$ and $\alpha_{ic}$ are defined as above; $R_{is}$ and $R_{ic}$ are the Inverse Mill Rations for solid fuel users and non-traditional fuel users, respectively; $\sigma_{\mu s}$ and $\sigma_{\mu c}$ are the covariance terms, defined as $\sigma_{\mu s}=\mathrm{cov}(\mu_{i},\varepsilon_{is})$ and $\sigma_{\mu s}=\mathrm{cov}(\mu_{i},\varepsilon_{ic})$, respectively; $\theta_{is}$ and $\theta_{ic}$ are error terms. We use the FIML approach to estimate the Equations (1), (2a) and (2b) simultaneously. If the correlation coefficients $\rho_{\mu s}=\frac{\sigma_{\mu s}}{\sigma_{\mu }\sigma_{s}}$ and $\rho_{\mu c}=\frac{\sigma_{\mu c}}{\sigma_{\mu }\sigma_{c}}$ are significantly different from zero, we may confirm that the unobservable factors would lead to the endogenous issue.

Based on the estimators from Equations (1), (2a) and (2b), we can calculate the treatment effects on the treated (ATT) which is the treatment effect of non-traditional fuel choices on depression. In addition, the ESR and ESP models allow for an overlap of $Z_{i}$ in Equation (1), and $X_{is}$ and $X_{ic}$ in Equations (2a) and (2b). However, at least one variable which acts as an instrument variable in Equation (1) but not in Equations (2a) and (2b) should be included for identification purposes. We use the ratio of non-traditional fuel use for other households, excluding the household in the same village as an instrument variable because it influences fuel type choice due to bandwagon effect but is irrelevant with the unobservable factors (e.g., the innate preference for clean and neat living conditions).

## 3. Results and Discussion

### 3.1. The Determinants of Households’ Non-Traditional Fuel Choices

Probit estimation results in Table 3 reveal that resident demographical characteristics, health conditions and economic situations significantly influence rural households’ decisions to adopt non-traditional fuels. In particular, age has a significantly negative effect on whether one uses non-traditional fuels, implying older residents are more likely to use solid fuels. Rural residents with religious beliefs tend to adopt solid fuels. Rural residents’ self-evaluation of health has a significantly positive impact on using non-traditional fuels. Compared to those who consider themselves in poor health, the probability of non-traditional fuel adoption is higher for people who consider themselves in better health. In our opinion, people who evaluate themselves in good health might care more about their health so they might pay more attention to factors which may influence their health, including cooking fuels. Household net worth per capita has a significant positive effect on using non-traditional fuels.

For the instrumental variable, the estimation results show that it has a significant positive effect on residents’ decision to use non-traditional fuels. Consequently, the instrumental variable can be valid. In Table 3, the mean ratio of non-traditional fuel use of other individuals in the same village is significant at the 1% statistical level, and the estimated coefficient is positive. This indicates that, to a certain extent, the greater the number of other residents in the same village using non-traditional fuels, the more likely it is that residents within the village will choose to use non-traditional fuels.

### 3.2. Impacts of Non-Traditional Fuel Choices on Depression

The results of the second stage of the ESR model regarding the impacts of different factors on depression are reported in the second and third columns of Table 3 for clean and solid fuel users, respectively. The impacts of control variables on depression are as follows. Age has a significant positive effect on depression among individuals using traditional solid fuels. The gender of the inhabitants has a negative effect on their depression. Women are more likely to be depressed, and the effect on their depression for non-traditional fuel users was more significant than for other individuals using solid fuels. Some studies have shown that women are more likely to be depressed than men [35,36].

Being religious is positive and significantly different from zero for non-traditional fuel users, indicating that religious people are more likely to be depressed, but the effect of being religious is not statistically significant for solid fuel users. Rural areas are more isolated, and residents may rely more on religions for their psychological and emotional needs. Furthermore, residents who use non-traditional fuels may have more needs and higher expectations from their religion, and this may result in depression if not achieved. Communist Party membership significantly affects the level of depression among solid fuel users, and communists are less likely to suffer from depression.

Both self-reported health status and hospitalization due to illness significantly influence depression for clean and solid fuel users. In addition, smoking has a significant positive effect on depression levels among solid fuel users. Smokers are more likely to be depressed, probably because they are exposed to more air pollutants from smoking, which tend to enhance the probability of depression.

In terms of economic circumstances, medical insurance has a negative effect on depression for solid fuel users, indicating that residents with medical insurance are less likely to suffer from depression. Residents with health insurance are able to cope with their health problems timely at lower costs, which is why they are less likely to suffer from depression. In addition, the depressive effect of household net worth per capita is negative and statistically significant at a 1% confidence level for both clean and solid fuel users.

The correlation coefficients $\rho_{1}$ and $\rho_{0}$ are both positive and statistically significant, suggesting that individuals who use non-traditional fuels (solid fuels) are less likely to suffer from depression than a random individual from the sample.

We also use the ESP model to estimate the impacts of non-traditional fuel choices on depression by replacing the continuous variable of depression level with a discrete variable. We calculate the results of marginal effects. The results are reported in Table 4 and are consistent with those by employing the ESR model.

The average treatment effects on treated using the ESR and ESP models are presented in Table 5. The results show that non-traditional fuel users have significantly lower Epidemiologic Studies Depression Scale (CES-D) scores and a lower probability of depression than solid fuel users, indicating that non-traditional fuel users face a lower risk of depression. In particular, adopting non-traditional fuels will lead to a 3.659 reduction in depression scores and decrease the probability of depression by 8.2%.

### 3.3. Heterogeneity Analysis

Compared to residents who use solid fuels, non-traditional fuel users had a significantly lower depression level for both males and females (see Table 6 below). However, the attenuating effect of non-traditional energy use on depression was more significant in females, possibly because females spend more time in the kitchen cooking and have more prolonged exposure to fumes than males and are therefore more susceptible to the effects of air pollution and even depression.

Regardless of self-reported social status, replacing solid fuels with non-traditional fuels can contribute to a lower depression score (see Table 6 below). However, the reduction effect was greater for high self-rated social statuses and for groups with very low self-rated social statuses than that for medium self-rated social statuses. In addition, compared to those solid fuel users, the use of non-traditional energy had a significant negative effect on depression for both those with and without health insurance. The extent of using non-traditional fuels to reduce depression is larger for the individuals who have no medical insurance.

### 3.4. Mechanism Analysis

We further investigate the mechanism underlying the association between non-traditional fuel choices and depression. The estimations using the ESR model are reported in Table 7. The results reveal that non-traditional fuel choices can significantly reduce the probability of physical discomfort and chronic disease in residents in rural China. Using solid fuels may result in a decline in air quality, and poor air quality may make people uncomfortable and even contribute to chronic disease. If this is not dealt with in a timely manner, prolonged exposure to pollutants can finally lead to depression. Therefore, replacing solid fuels with non-traditional fuels will contribute to a lower risk of physical discomfort and chronic disease, and further result in lower depression levels.

## 4. Discussion

The empirical results show that employing non-traditional fuels can decrease the probability of depression by 8.2%. The prevalence of depression is 35% in our sample, using the cutoff point of 16 to define depressive symptoms. Based on the empirical results, we may conclude that replacing solid fuels with non-traditional fuels will decrease the prevalence of depression to 32.1% on average. It implies that about 717 residents in our sample will eliminate depression risk by using non-traditional fuels. Moreover, these residents will be happier in their future lives, and have a much lower burden on their families, society, and medical institutions. Using non-traditional fuels will also reduce the risk of physical discomfort and chronic disease for residents in rural China. The Chinese government attached great attention to rural environmental protection and implemented a coal-to-gas policy in 2017 to promote rural energy transition.

In addition, the findings in this paper are consistent with some other articles. First, we find that the older those who used firewood and coal were, the more likely they were to suffer from depression. Older rural residents are more likely to be depressed because they are more accustomed to traditional cooking methods and have been exposed to pollutants from fuels for extended periods. The result is consistent with the findings in [36] and [1]. Second, some studies documented that depression may arise from physical illness, including chronic disease [1,36]. This paper reveals the relationship between health conditions and depression. We find that self-reported health status and hospitalization have a more significant impact on depression for the group of individuals who used solid fuels. One possible explanation is that using solid fuels for cooking exposes residents to more pollutants. It may increase not only the probability of depression but also the probability of contracting other cardiopulmonary diseases than those who use non-traditional fuels. Last but not least, this paper finds that residents with higher household net worth per capita are less likely to use solid fuels, probably because these households are better off and therefore have more money to use gas and buy appliances such as hoods. This is consistent with the result that households’ economic status is a crucial determinant of fuel use in urban Ethiopia [43]. Although the Chinese government has implemented a series of policies to promote rural energy transition from solid fuel use to non-traditional fuel use, we should pay more attention to the economic conditions of residents and reduce the cost of using non-traditional fuels for poor households.

## 5. Conclusions

We have seen a dramatic increase in public concerns over depression in recent years, and we lack effective therapies to treat depression. Therefore, it is essential to study the determinants of depression and to avoid the risk of depression. A large number of studies have explored the effects of outdoor air pollution on depression. Since solid fuels such as charcoal, wood, and crop residues are widely used and will contribute to air pollution in the house in rural China, this paper aims to examine whether and to what extent non-traditional fuel adoption reduces depression. Based on CFPS datasets, this paper employs the ESR and ESP models to investigate the impact of non-traditional fuel use on depression. The empirical results show that non-traditional fuel users have significantly lower Epidemiologic Studies Depression Scale (CES-D) scores, indicating non-traditional fuel users face a lower risk of depression. Compared to solid fuels, employing non-traditional fuels will lead to a 3.659 reduction in depression score or decrease the probability of depression by 8.2%. In addition, the results of the mechanism analysis show that household non-traditional fuel choices affect depression by reducing the probability of physical discomfort and chronic disease.

This paper has policy implications. China’s Government, as well as governments in developing countries, need to vigorously promote the use of non-traditional fuels, which is not only good for improving the quality of environment, but also good for the mental health of residents.

This paper has limitations. This paper only employs the data in 2012 because all of the individuals were surveyed using 20 items to measure the CES-D scores. After 2012, 80% of the individuals were tested by using eight of twenty items every year, and depression levels cannot be compared across years. The mechanism analysis in this paper is preliminary due to data limitations and can be deepened in the future. In addition, many of the depression scales were developed in females and are more geared towards signs and symptoms of depression in women, which might differ from signs and symptoms in men. Thus, the depression score between male and female may be incomparable due to measurement error.

## Figures and Tables

**Table 1 ijerph-19-15639-t001:** Variable measurements and descriptive statistics of variables.

Variables	Definition	Mean	S.D
Dependent variables			
Depression level	The sum scores of 20 items ranging from 0 to 60 and one of the items takes the score: 0 if rarely means less than one day for a week; 1 if some of the time means 1–2 days for a week; 2 if occasionally means 3–4 days for a week; 3 if most of the time means 5–7 days for a week;	13.64	8.18
Depression status	The cutoff point of 16 or more is used to classify patients with depressive symptoms: 1 if the depression level larger is than 16, 0 if otherwise	0.35	0.48
Key explanatory variables			
Non-traditional fuel adoption	1 if use non-traditional fuels including liquefied gas, natural gas, and electricity, and 0 for using solid fuels including coal, charcoal, wood, crop residues	0.49	0.50
IV	The ratio of solid fuel use for other households excluding the household in the same village	0.48	0.33
Control variables			
Age	Age of the individual (years)	45.85	16.64
Sex	Sex of the individual: 1 if male, 0 if otherwise	0.48	0.50
Religion	1 if the individual has religion, 0 if otherwise	0.12	0.33
Ethnicity	1 if the ethnicity of individual is other, 0 for Han	0.17	0.37
CPC membership	1 if the individual is a member the Communist Party of China, 0 if otherwise	0.05	0.22
Health status	Health status assessed by investigators: from 1 = very bad to 7 = very good	5.31	1.14
Uncomfortable	1 If the individual felt uncomfortable in past two weeks, 0 if otherwise	0.31	0.46
Chronic disease	1 if the individual had a chronic disease in past six months, 0 if otherwise	0.12	0.32
Hospital	1 if the individual visited hospitalized in past year, 0 if otherwise	0.09	0.29
Smoking	1 if the individual smoked in past month, 0 if otherwise	0.30	0.46
Medical insurance	1 if the individual has medical insurance, 0 if otherwise	0.91	0.29
Asset per capita	The households’ net asset per capita (10,000 Yuan)	6.72	14.97
Family size	Total number of people residing in a household (persons)	4.57	1.91
Exercise	1 if almost every day; 2 if two or three times a week; 3 if two or three times a month; 4 if once a month; 5 if never	3.66	1.75

**Table 2 ijerph-19-15639-t002:** Mean differences in characteristics between solid fuel users and non-traditional fuel users.

Variables	Non-Traditional Fuel Users	Solid Fuel Users	Differences
Depression level	12.229 (0.073)	14.970 (0.081)	−2.741 ***
Depression status	0.280 (0.004)	0.423 (0.005)	−0.143 ***
Age	44.418 (0.159)	47.197 (0.158)	−2.779 ***
Sex	0.484 (0.005)	0.485 (0.005)	−0.001
Religion	0.126 (0.003)	0.116 (0.003)	0.009 **
Ethnicity	0.179 (0.004)	0.159 (0.003)	0.019 ***
CPC membership	0.054 (0.002)	0.048 (0.002)	0.005 **
Health status	5.449 (0.010)	5.180 (0.011)	0.269 ***
Uncomfortable	0.280 (0.004)	0.344 (0.004)	−0.064 ***
Chronic disease	0.109 (0.003)	0.129 (0.003)	−0.021 ***
Hospital	0.085 (0.003)	0.097 (0.003)	−0.012 ***
Smoking	0.282 (0.004)	0.312 (0.004)	−0.029 ***
Medical insurance	0.894 (0.003)	0.916 (0.003)	−0.023 ***
Asset per capita (10,000 Yuan)	9.587 (0.188)	4.044 (0.083)	5.544 ***
Family size	4.494 (0.018)	4.640 (0.018)	−0.147 ***
Exercise	3.586 (0.017)	3.720 (0.017)	−0.134 ***

Note: ** *p* < 0.05 and *** *p* < 0.01. Standard deviation is presented in parentheses.

**Table 3 ijerph-19-15639-t003:** The coefficients and standard errors of ESR model for the effect of non-traditional fuel choices on depression level.

Variables	Select Function	Depression Level	
		Non-Traditional Fuel Users	Solid Fuel Users
Age	−0.002 (0.000) ***	0.036 (0.005) ***	0.068 (0.005) ***
Gender	0.003(0.006)	−2.017 (0.179) ***	−2.249 (0.201) ***
Religion	−0.019 (0.008) **	0.846 (0.236) ***	0.685 (0.257) ***
Ethnicity	−0.002 (0.007)	0.100 (0.192)	−0.210 (0.211)
CPC membership	0.032 (0.012) ***	−1.172 (0.284) ***	−2.059 (0.331) ***
Health status	0.019 (0.002) ***	−0.580 (0.072) ***	−0.862 (0.069) ***
Hospital	0.001 (0.009)	2.642 (0.297) ***	2.677 (0.287) ***
Smoking	−0.009 (0.007)	0.135 (0.193)	0.392 (0.214) *
Medical insurance	0.006 (0.009)	−0.131 (0.239)	−0.679 (0.289) **
Asset per capita	0.002 (0.001) ***	−0.022 (0.004) ***	−0.050 (0.011) ***
Family size	−0.002 (0.001)	−0.003 (0.039)	−0.038 (0.042)
Exercise	−0.000 (0.001)	0.205 (0.042) ***	0.209 (0.045) ***
IV	0.076 (0.004) ***		
Constant	——	13.336 (0.574) ***	18.024 (0.604) ***
$\rho_{1}$		0.197 (0.026) ***	
$\rho_{0}$			0.215 (0.019) ***
Wald test	169.86 *** (*p*-value = 0.000)		
Observations	21,324	21,324	21,324

Note: * *p* < 0.1, ** *p* < 0.05 and *** *p* < 0.01. Robust standard deviation is presented in parentheses.

**Table 4 ijerph-19-15639-t004:** The coefficients and standard errors of ESP model for the effect of non-traditional fuel choices on depression status.

Variables	Select Function	Depression Status	
		Non-Traditional Fuel Users	Solid Fuel Users
Age	−0.009 (0.001) ***	0.004 (0.001) ***	0.008 (0.001) ***
Gender	0.012 (0.027)	−0.327 (0.034) ***	−0.286 (0.031) ***
Religion	−0.079 (0.033) **	0.160 (0.040) ***	0.050 (0.039)
Ethnicity	−0.010 (0.029)	0.002 (0.036)	−0.068 (0.034) **
CPC membership	0.138 (0.050) ***	−0.211 (0.066) ***	−0.221 (0.060) ***
Health status	0.079 (0.009) ***	−0.069 (0.013) ***	−0.129 (0.010) ***
Hospital	0.001 (0.037)	0.409 (0.045) ***	0.369 (0.041) ***
Smoking	−0.036 (0.029)	0.064 (0.038) *	0.052 (0.034)
Medical insurance	0.023 (0.037)	−0.018 (0.044)	−0.108 (0.044) **
Asset per capita	0.010 (0.002) ***	−0.004 (0.001) ***	−0.008 (0.003) ***
Family size	−0.010 (0.006) *	−0.014 (0.007) **	0.003 (0.006)
Exercise	−0.000 (0.006)	0.023 (0.008) ***	0.017 (0.007) **
IV	0.195 (0.005) ***		
Constant	——	−0.365 (0.104) ***	0.377 (0.094) ***
$\rho_{1}$		0.222 (0.031) ***	
$\rho_{0}$			0.273 (0.027) ***
Wald test	136.81 *** (*p*-value = 0.000)		
Observations	21,324	21,324	21,324

Note: * *p* < 0.1, ** *p* < 0.05 and *** *p* < 0.01. Robust standard deviation is presented in parentheses. The estimators reported in Table 4 are margin effects.

**Table 5 ijerph-19-15639-t005:** Average treatment effects on treated using ESR and ESP model.

Outcome	ESR Model		Outcome	ESP Model	
	ATT	*t*-Value		ATT	*t*-Value
Depression level	−3.659 (0.009) ***	−4.1 × 10^2^	Depression status (0/1)	−0.082 (0.002) ***	−46.471

Note: *** *p* < 0.01. Robust standard deviation is presented in parentheses. ATT refers to average treatment effects on treated.

**Table 6 ijerph-19-15639-t006:** The average treatment effects on treated of heterogeneity analysis.

Variables	ATT	*t*-Value
Panel A: Gender heterogeneity		
Male	−3.546 (0.010) ***	−3.6 × 10^2^
Female	−3.857 (0.010) ***	−4.2 × 10^2^
Panel B: Social status heterogeneity		
Self-reported social status is low	−3.379 (0.012) ***	−2.7 × 10^2^
Self-reported social status is medium	−2.955 (0.012) ***	−2.4 × 10^2^
Self-reported social status is high	−5.477 (0.016) ***	−3.3 × 10^2^
Panel C: Formal social support heterogeneity		
With medical insurance	−3.386 (0.009) ***	−3.8 × 10^2^
Without medical insurance	−5.287 (0.023) ***	−2.3 × 10^2^

Note: *** *p* < 0.01. Robust standard deviation is presented in parentheses.

**Table 7 ijerph-19-15639-t007:** Average treatment effects on treated using the ESP model.

Outcome	ESP Model			ESP Model	
	ATT	*t*-Value		ATT	*t*-Value
Physical discomfort (0/1)	−0.042 (0.002) ***	−27.909	Chronic disease (0/1)	−0.014 (0.001) ***	−20.427

Note: *** *p* < 0.01. Robust standard deviation is presented in parentheses. ATT refers to average treatment effects on treated.

## Data Availability

The data presented in this study are available upon request from the corresponding author otayu@webmail.hzau.edu.cn (G.Y.).

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
