# Peer review of "Ecological Sustainability and Households’ Wellbeing: Linking Households’ Non-Traditional Fuel Choices with Reduced Depression in Rural China"

_ijerph, 2022, doi:10.3390/ijerph192315639_

Round 1

Reviewer 1 Report

The statistics of the manuscript do not provide a sound basis for the conclusions. Based on Table 2, solid fuel users (compared to clean fuel users) have a higher depression level, have a higher age, are not religious, belong to a lower extent to the CPC, have lower health status, suffer from chronic disease, and have lower income. What if depression is an outcome of their lower income? Lower-income people use solid fuels, as the literature says. Lower-income can yield to depression by itself, with all the social aspects that go along.

I think that there are a lot of confounding factors that yield to depression and trying to link solid fuel usage with it is not necessarily beneficial. Solid fuel use without proper aeration yields poor health due to air pollution and poor health leads to depression. That is the bottom line. 

Reviewer 2 Report

This is an interesting examination of the association of depression and choice of household fuel source in rural China.  Associations have been reported between pollution and depression and this study explores these connections within the household of those using natural gas and electricity rather than burning biomass sources.  There are statistical concerns.  Referring to these more modern fuel alternatives as “clean” is problematic.  They have been not shown to be cleaner in the sense that they are polluting since they leak methane.  Gas stoves continually leak methane at the household level and the pipes which carry natural gas also leak methane.  Coal is not clean and electricity is not clean if it comes from coal or natural gas.  A better term for these fuel sources might be a good choice to no promote the false idea that they are clean.  They are still fossil fuels. Maybe calling them “non-traditional” would work in the rural Chinese context.

The use of endogenous switching regression from econometrics is an interesting approach to account for selection bias. This is an instrumental variable analysis and finding a good instrumental variable can be challenging.  This is a well-accepted approach to adjusting for unobserved confounding variables.

Editing for English is needed, but overall it is fairly well written.

Questions and comments:

1.  The CES-D scale is not known to be normally distributed. In fact, it is almost always positively skewed and cannot be transformed to normality, yet in the study it was used as a continuous variable.  The scale summary variable, the sum score, is nearly always dichotomized at 16.  There was no information provided on the distribution of the CES-D variable as a continuous score in the methods section. The probit model is probably a better choice than the linear model. Significance is not surprising given the sample size.  It might be best to select the most appropriate model for the data and go with only one model.  Without knowing whether the continuous CES-D score was even close to normal, the probit model is more appropriate since it does not need to meet model assumptions.  Even though this is an instrument variable analysis, the ESR is still assuming normality.

2.  Is access to more modern fuels not known in China?  It would seem that this information would be available and could be mapped.  For example, it seems as if the location of an electric grid would be available.  Also, it should be known where underground natural gas pipelines are located.  This is hard to believe, but maybe it is true.  Maybe it is also true that even if these more modern forms of energy are available, people opt not to use them due to cost and wanting to adhere to traditional ways.  Also, maybe because it would mean purchasing new appliances to use these types of energy such as stoves and ovens rather than cooking over wood and other forms of biomass.

3.  It is true that a higher CES-D score means a higher risk of depression, and the manuscript clearly states this, but in lines 159-161 it mistakenly says the opposite and should be corrected.  It looks like it should say that “clean” energy users face a lower risk of depression.

4.  The results shown in Table 2 tend to make the case that those who choose modern fuels to biomass-type fuels are clearly different, however, you have a very large sample size.  It is not surprising that small effects would become statistically significant.  Have you considered standardizing your p-value to adjust for the large sample size using the method of Good, 1982.  Some of these significant differences are quite small and it is an interesting question as to whether they are truly relevant.

5.  The instrumental variable you are using, the ratio of villagers using modern fuels, seems to be a surrogate for the availability of these modern fuel types in a village.  It makes sense that availability would have a minimal effect on an individual’s depression level if all subjects are in rural areas.  Is it true that these areas with access to modern fuels are random among rural areas or is there a reason the government provides some rural areas with alternatives to traditional biomass and others are left without access?

6.  Nice job explaining the regression methods.  It would be good to put headings in Tables 3 and 4 describing that they ae regression coefficients and standard errors.

7. It is a bit odd to see means calculated for binary variables.  It might be good to show the frequencies rather than the means, or maybe show both if you wish to keep the means.

8. Although it is possible that women are more likely to score as being more depressed, there are a number of reasons for this.  Many of the depression scales were developed in females and are more geared towards signs and symptoms of depression in women, which might differ from signs and symptoms in men.  It might be that we better measure depression in women than in men.  More work should be done on this.  I think to say that women are more depressed ignores all the issues that come with measurement error in the depression scales commonly used.

9.  Lines 320-321 – Sentence is not clear.

10. Line 42 also needs editing.

11. Should mention that this is cross-sectional data and causality cannot be assumed.

12.  It seems that it is possible that the use of traditional fuels could be a surrogate variable for other factors or other environmental exposures that would be closely related to fuel choice, which could also contribute to depression.  This should also be discussed so that undue causal weight is not given to the indoor air pollution due to fuel choices.

Round 2

Reviewer 1 Report

All my comments have been answered